# Unrealized fertility among women in low and middle-income countries

**Shireen Assaf**[1,2]*, **Lwendo Moonzwe Davis**[2]

**1** The DHS Program, Rockville, Maryland, United States of America, **2** ICF, Rockville, Maryland, United States of America

* shireen.assaf@icf.com

**Data Availability Statement:** Data are publicly available on The DHS Program website after a simple registration: https://www.dhsprogram.com/data/available-datasets.cfm.

## Abstract

### Background

There has been little research on women who have fewer than their ideal number of children toward the end of their childbearing years in low and middle-income countries (LMICs). We examine the level and distribution of unrealized fertility in LMICs across three geographical regions. We also examine the role of sex preference and other factors associated with unrealized fertility.

### Data and methods

We used Demographic and Health Survey (DHS) data for women age 44–48 in 36 countries from the three geographical regions of Western and Central Africa, Eastern and Southern Africa, and South and Southeast Asia. We conducted descriptive analysis to examine the distribution of unwanted fertility and unrealized fertility, and fit adjusted logistic regressions of unrealized fertility. The main variables are number of living children (including by sex) and the sex composition of children. Other variables included education, marital status, age at first childbirth, wealth quintile, place of residence, exposure to family planning messages, contraceptive use, and country.

### Results

Unrealized fertility was highest in Western and Central Africa, followed by Eastern and Southern Africa. In all regions, there was a decrease in unrealized fertility with an increasing number of children. Findings for sex preference varied with little sex preference in the African regions, and some limited evidence of preference for sons in South and Southeast Asia. In most regions, higher levels of education, higher wealth quintile, and use of contraceptive methods were associated with decreased unrealized fertility.

### Conclusion

Family planning programs and messages should consider regional and socioeconomic differences in unrealized fertility in order to give women and families the right to achieve the family size they desire regardless of their status.

**Funding:** This study was funded by the United States Agency for International Development (USAID) through ICF/The DHS Program.

**Competing interests:** The authors have declared that no competing interests exist.

## Introduction

Studying fertility preferences and fertility behaviors is important for planning programs that help women achieve the healthy family size that they desire. At the same time, family planning programs, especially in low and middle-income countries (LMICs), are concerned with high fertility levels and unintended pregnancies. High fertility is linked to negative effects on maternal and child health outcomes and mortality [1–3], and at the macro level, to social, economic, and environmental challenges [4]. Prior research in LMICs has focused on unwanted fertility and pregnancies in an effort to reduce projected fertility levels. Women with fewer children than their ideal number have unrealized fertility, while those with more than their ideal number have unwanted fertility. Since reproductive rights include the right to freely decide the number, spacing, and timing of children, it is important to study both unwanted fertility and unrealized fertility. Although there is increasing interest in studying unrealized fertility and the factors associated with women not meeting their fertility goals, little research has been done in this area in LMICs [5]. Understanding these desires and associated factors can better inform planning programs to help women achieve fertility ideals, while supporting their health and the health of their families.

Although the global fertility rate declined from 3.2 to 2.5 live births per woman between 1990 and 2019, the decline is occurring at a slower rate in Sub-Saharan Africa (SSA), where fertility rates remain high [6]. In SSA, the estimated fertility rate was 4.6 births per woman as of 2019, as compared to 1.8 in Eastern and South Eastern Asia [6]. Despite the high fertility rates in SSA, unrealized fertility remains the highest in SSA compared to other regions [5, 7]. This indicates that the relationship between unrealized fertility and fertility is not a simple inverse relationship.

One study that used data from 227 Demographic and Health Surveys (DHS) in 58 countries found that 40% of women in SSA had unrealized fertility, compared to 26% in non-SSA countries [7]. In addition, while one study with data from 30 SSA countries found that a woman's level of education was negatively associated with increased fertility and desired family size [8], higher education was not found to correspond with lower unrealized fertility in SSA [7]. One of the few country-specific studies on unrealized fertility in LMICs found that unrealized fertility in Nigeria was associated with ethnicity and sex composition [9].

Women's fertility desires play an important role in behavior. A synthesis of longitudinal studies conducted across Africa and Asia found that women's desire to stop childbearing was strong predictor of their subsequent fertility, and that women who did not want any more children were less likely to have children [10]. However, particularly in SSA, the study found a discrepancy between desire to stop having children and behavior, as well as greater uncertainty and instability in childbearing preferences for women in SSA [10]. Instability and uncertainty may also be affected by external factors. For example, in low income agricultural societies, there is a tendency to desire larger numbers of children compared to higher income societies with more developed secondary and tertiary sectors [4].

Sex preference may also be a key factor in unrealized fertility intention. However, the findings from studies that examined the relationship between sex preference and fertility intentions have been inconsistent. Associations also vary by country and regions within a country. One study conducted in 50 countries found that a balance between daughters and sons is common, although in some Latin American, Caribbean, and several Southeast Asian countries, there was a preference for daughters, while in Southern and Western Asian and Northern African countries, there tended to be a preference for sons [11, 12]. For SSA countries, a son preference was found in 16 of the 28 countries [11]. Another study in Nigeria also found that having only daughters was associated with higher odds of unrealized fertility [9]. Sex

preference may also play a role in fertility intention. For example, another study found a larger fertility response to the absence of sons in Central and South Asia [13].

There is a need for research to understand the factors associated with unrealized fertility, especially in LMICs. This paper examines unrealized fertility in women age 44–48. We focus on this age group because we assume that women in this age group are near the end of their childbearing years, and are suitable for comparing fertility preferences to the number of living children. Casterline and Han (2017), who examined two measures of unrealized fertility—-lower bound (desire for another child) and upper bound (ideal greater than current number of children)—have found unrealized fertility to be more prevalent in the upper bound compared to the lower bound [5]. One important limitation of the desire for another child measure is that the measure does not include all women (infecund, sterilized, and undecided). This paper examines the level and distribution of unrealized fertility in three geographical regions using the upper bound measure that focuses on measuring if women are falling short of their desired number of children at a specific point in time. We examined the factors associated with unrealized fertility, such as the number of children, sex preference, and sociodemographic variables. We expect women with fewer children, lower education levels, lower wealth status and those who are not using contraception to have higher unrealized fertility. We also attempted to examine sex composition in relation to unrealized fertility and if having more boys or girls is an important driver. Some country-specific findings are discussed, and all are available in the S1–S3 Tables.

## Data and methods

### Data

Data from 36 countries with a DHS conducted since 2014 were included in the analysis (see Table 1). The countries were grouped into three geographical regions: Western and Central Africa (13 countries), Eastern and Southern Africa (11 countries), and South and Southeast Asia (12 countries).

### Measures

We focus on two variables in this analysis: the ideal number of children and the number of living children. In the DHS, the question for ideal number of children is asked differently depending on if the woman has any children or not. For women with living children, the question asks:

> "If you could go back to the time you did not have any children and could choose exactly the number of children to have in your whole life, how many would that be?"

Women without living children are asked:

> "If you could choose exactly the number of children to have in your whole life, how many would that be?"

This question is followed by a question about how many of the children the mother would like to be boys, girls, or either. This is the information on the women's ideal number of children, ideal number of boys, and ideal number of girls. A non-numerical response such as "It is up to God" can range from 0–29% among women age 44–49 depending on the country. The majority of countries (26 of 36) had non-numeric responses less than 15%. Women who

**Table 1. Surveys included in the analysis.**

| Region | Country | DHS surveys |
|---|---|---|
| Western and Central Africa | Angola | 2015–16 |
| | Benin | 2017–18 |
| | Chad | 2014–15 |
| | Cameroon | 2018–19 |
| | Gambia | 2019–20 |
| | Ghana | 2014 |
| | Guinea | 2018 |
| | Liberia | 2019–20 |
| | Mali | 2018 |
| | Mauritania | 2019–21 |
| | Nigeria | 2018 |
| | Senegal | 2019 |
| | Sierra Leone | 2019 |
| Eastern and Southern Africa | Burundi | 2016 |
| | Ethiopia | 2016 |
| | Kenya | 2014 |
| | Lesotho | 2014 |
| | Malawi | 2015–16 |
| | Rwanda | 2019–20 |
| | South Africa | 2016 |
| | Tanzania | 2015–16 |
| | Uganda | 2016 |
| | Zambia | 2018–19 |
| | Zimbabwe | 2015 |
| South and Southeast Asia | Afghanistan | 2015 |
| | Bangladesh | 2017–18 |
| | Cambodia | 2014 |
| | India | 2019–20 |
| | Indonesia | 2017 |
| | Maldives | 2016–17 |
| | Myanmar | 2015–16 |
| | Nepal | 2016 |
| | Pakistan | 2017–18 |
| | Papua New Guinea | 2016–18 |
| | Philippines | 2017 |
| | Timor-Leste | 2016 |

provided a non-numeric response were excluded from the analysis. This is a limitation of our analysis that would disproportionally affect some regions more than others.

By taking the difference of the ideal number of children and the current number of living children, we constructed a variable categorized as:

1. unwanted fertility (ideal is less than current number)

2. ideal equals current number, and

3. unrealized fertility (ideal is greater than the current number).

As shown in the S1 Table, the percentage of women who had a birth between age 44–48 is approximately 1% or less for all countries. Therefore, the number of living children is relatively

low for this age group and is appropriate for the study of unrealized fertility. Women age 49 were excluded from the analysis to reduce bias from age displacement in survey responses [5].

Similar variables were constructed for the descriptive results by taking the difference between the ideal number of boys and current number of living sons, and the difference between the ideal number of girls and current number of living daughters. The comparison of the ideal number of children by sex with the total ideal number provided an initial indication of sex preferences.

## Methods

We used descriptive results to examine the distribution of unwanted fertility and unrealized fertility for each region. This was performed for the total number of children and for each sex. The focus of the paper, however, is unrealized fertility. To examine this further, we fit adjusted logistic regressions of unrealized fertility for the total number of children. Therefore, we combined unwanted fertility and the ideal to produce a binary variable for unrealized fertility versus no unrealized fertility.

The main variables of interest are the fertility related variables: number of living children, number of living boys, number of living girls, and sex composition. Due to the different distributions of these variables by regions, these variables were categorized differently depending on the region, as shown in Table 2. Exploration of the sex composition of children by regions determined the categories in Table 2. For example, categories that represent extreme distributions, such as only sons and only daughters, were not constructed due to small sample sizes in these categories. The fertility variables were highly correlated and could not be entered into the same regression model. Therefore, a separate model was fit for each variable and each region. The exception was the number of living boys and number of living girls that were not highly correlated and could be entered in the same model (Model 2).

Other variables examined include: education level (none, primary, secondary or more), marital status (never in a union, currently in a union, and formerly in a union, which combined widowed, separated and divorced women), age at first birth (less than 20, 20–24, 25–39, 30–49), wealth quintile, place of residence, exposure to family planning (FP) messages from radio, television, or newspapers (exposed or not exposed), contraceptive use (none, traditional, modern method), and country. These variables have been found in the literature to have direct links to fertility preferences and behaviors.

We use the country variable to account for between country variability. The country with the lowest percentage of unrealized fertility in the descriptive results was the reference country in the logistic regression model. All analyses considered the sampling design and sampling weights for each country. To give equal weight to the surveys in the analysis, we multiply the

**Table 2. Coding of fertility variables by region.**

| Variable | African regions | Remaining regions | Model |
|---|---|---|---|
| Number of children | 0–2, 3, 4, 5, and 6 or more children | 0–1, 2, 3, and 4 or more children | 1 |
| Number of living boys or girls | 0–1, 2, 3, and 4 or more | 0, 1, 2, and 3 or more | 2 |
| Sex composition | • 0–3 children,<br>• 4 children with more sons than daughters,<br>• 4 children with more daughters than sons,<br>• 2 sons and 2 daughters,<br>• 5 children with more sons than daughters,<br>• 5 children with more daughters than sons, and<br>• 6 or more children | • 0–1 children,<br>• 1 son and 1 daughter,<br>• 2 sons and no daughters,<br>• no sons and 2 daughters,<br>• 3 children with sons more than daughters,<br>• 3 children with daughters more than sons, and<br>• 4 or more children | 3 |

sample weight by the total weight for all countries, divided by 43 surveys, and then divided by the observed weight for each country. A unique strata and primary sampling unit was constructed for each survey. The *svy* command for complex survey data was used for all analyses with Stata 17 software.

## Results

### Ideal number of children

Fig 1 summarizes the mean ideal number among women age 44–48 for all countries in the analysis. The estimates are also shown in S2 Table, along with the distribution of ideal number of children and the percentage of non-numeric responses. In general, there is preference for larger family sizes in the African regions compared to South and Southeast Asia. The highest

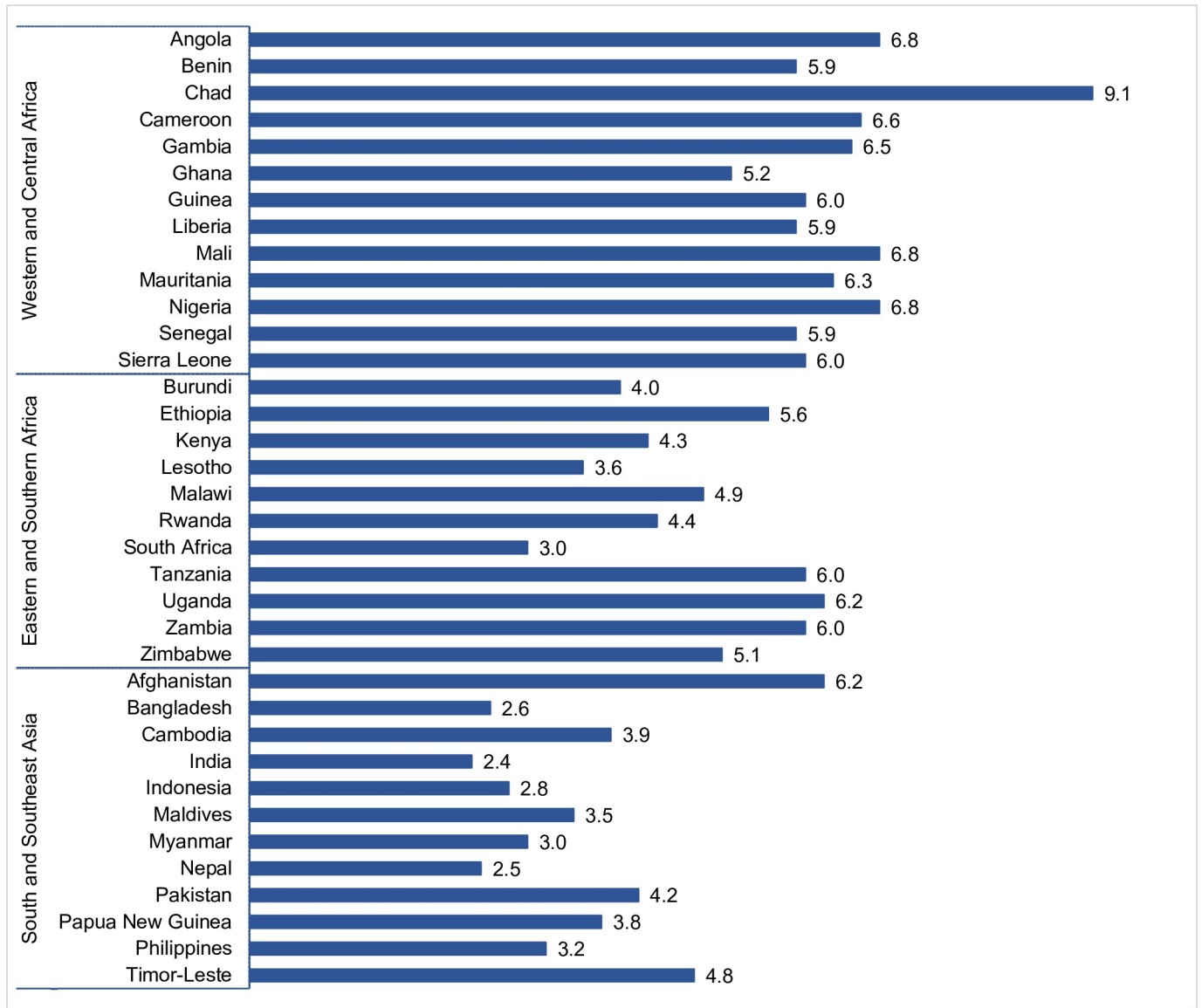

**Fig 1. Mean ideal number of children among women age 44–48.**

means were found in Western and Central Africa. There were some countries that deviated from the regional trend. For example, the mean ideal number of children in Chad was 9.1, compared to 5.1 to 6.8 in the remaining countries. Afghanistan also had a higher mean (6.2) compared to other countries in South and Southeast Asia (between 2.4 and 4.8). Several countries in Western and Central African had high percentages of non-numeric responses, with six of 13 countries above 15%. The highest was in Chad where 29% of women age 44–48 gave a non-numeric response. In Eastern and Southern Africa, less than 5% of women age 44–48 from most countries gave non-numeric responses, while in Ethiopia, 19% of women gave this response. We also find a high percentage of non-numeric responses in Afghanistan (25%) compared to the remaining countries in that region. Several countries, including Angola, India, Lesotho, Nepal, the Philippines, and South Africa, had a non-numeric response rate less than 1%. Non-numeric responses were excluded from the remaining analysis in the report.

## Distribution of unwanted fertility and unrealized fertility

We took the difference between the ideal number of children and the current number of living children, and observed where women age 44–48 have experienced unwanted fertility (ideal<current number), have reached their ideal (ideal = current number), or have unrealized fertility (ideal>current number). Figs 2 to 4 summarize this distribution in each region for

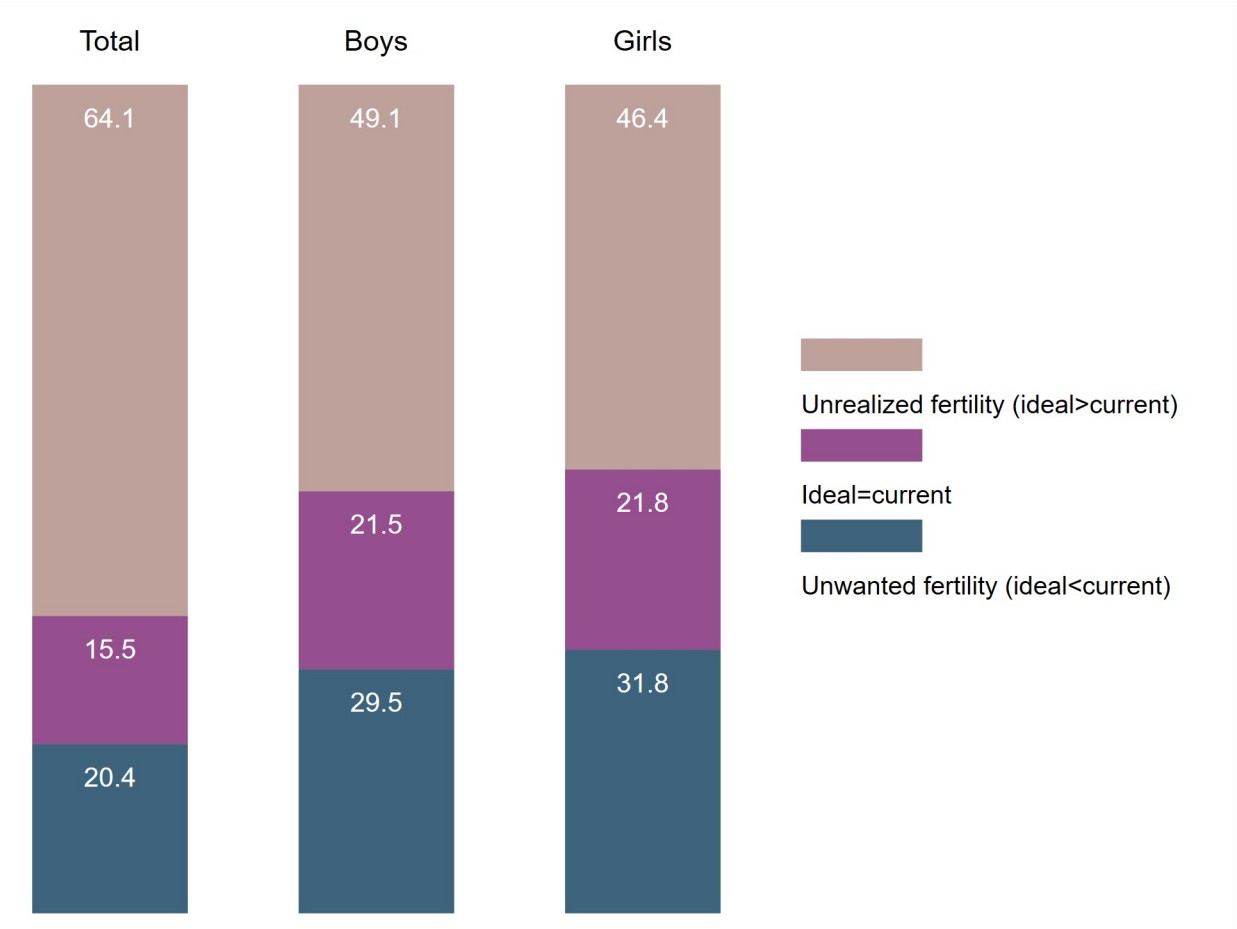

**Fig 2. Distribution of ideal minus current number of children among women age 44–48, Western and Central Africa.**

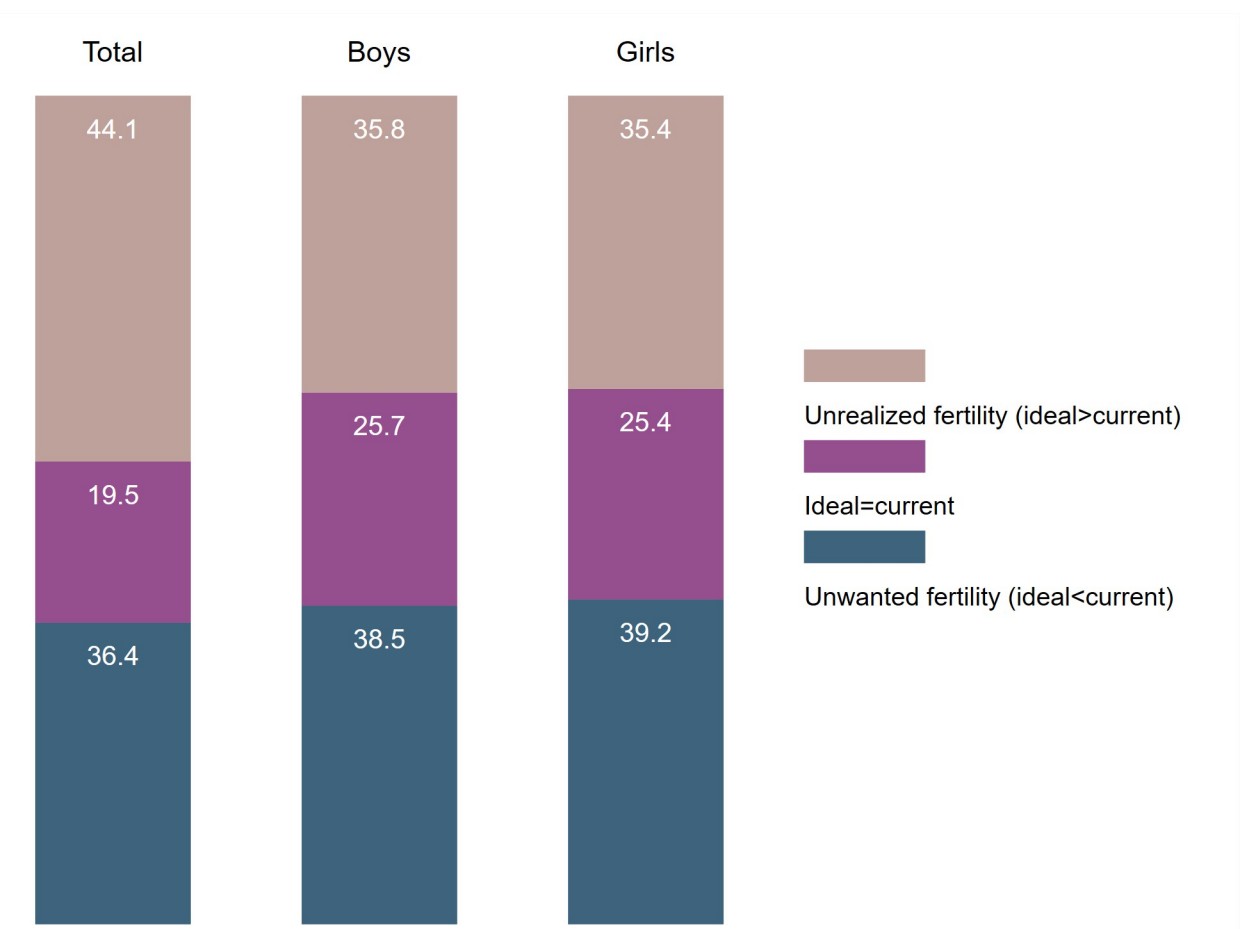

**Fig 3. Distribution of ideal minus current number of children among women age 44–48, Eastern and Southern Africa.**

total children, boys, and girls. The S3 Table provides these estimates with 95% confidence intervals (C.I.), as well as the country-specific estimates.

In the comparisons of the three regions in Figs 2 to 4, unrealized fertility among women age 44–48 is the highest in Western and Central Africa (64.1%), followed by Eastern and Southern Africa (44.1%), and South and Southeast Asia (31.0%). The country with the highest unrealized fertility was Chad with 75.8% of women with unrealized fertility and the lowest in Nepal with 11.8% of women with unrealized fertility (see S3 Table). Unwanted fertility was very similar for Eastern and Southern Africa and South and Southeast Asia (approximately 36%), but was lower in Western and Central Africa (20.4%). The South and Southeast Asia region had the highest percentage of women age 44–48 who reached their ideal (33.3%). This was mainly due to the high percentages of women in India and Indonesia who reached their ideal number (46.3% and 42.2%, respectively). Only 15.5% of women in Western and Central Africa had reached their ideal number of children.

Unrealized fertility for boys or girls was always lower than the total for all regions. This suggested that the number of children was more important than then sex when women reported their fertility preferences. In addition, the distribution of the difference between ideal number of boys and current number of sons, and ideal number of girls and current number of daughters were very similar and almost identical in Eastern and Southern Africa and South and

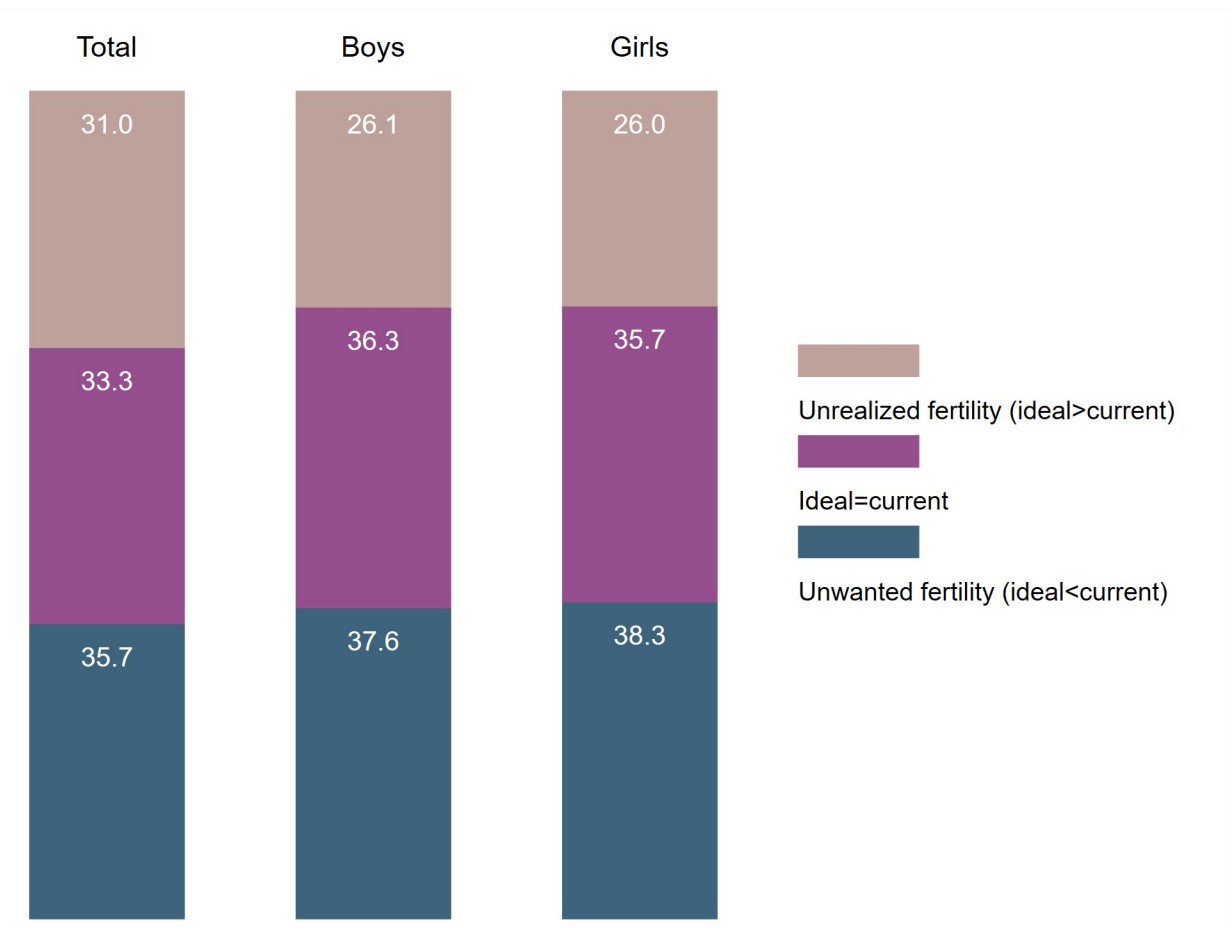

**Fig 4. Distribution of ideal minus current number of children among women age 44–48, South and Southeast Asia.**

Southeast Asia. In Western and Central Africa, we observe a slight preference for having boys; 49.1% of women age 44–48 had unrealized fertility for boys compared to 46.4% for girls.

## Crosstabulation of unrealized fertility with background variables

Table 3 summarizes the results of the crosstabulation of all independent variables with unrealized fertility. For all regions, we observe a decrease in unrealized fertility with increasing number of children, sons, and daughters. In the African regions, there appears to be little evidence of sex preferences. We observe that in these regions, the percentage of unrealized fertility among women who have 4 or 5 children did not change significantly according to the sex composition. While the percentages are different, the confidence intervals overlap. However, in South and Southeast Asia, there appears to be some son preference due to a higher percentage of unrealized fertility when there are more daughters than sons for women with three children (32.2%), compared to women with more sons than daughters (27.5%). However, the confidence intervals were slightly overlapping.

Unrealized fertility increased with increasing women's education level, except for women in Western and Central Africa. In Western and Central Africa, women with no education had higher unrealized fertility compared to women with primary or higher education. Women who were never or formerly in a union had higher unrealized fertility compared to women

**Table 3. Crosstabulation of unrealized fertility with background variables for each region.**

| Variable | Western and Central Africa | Eastern and Southern Africa | | South and Southeast Asia |
|---|---|---|---|---|
| Number of children | | | Number of children | |
| 0–2 | 92.3 [90.6,93.8] | 80.3 [77.9,82.4] | 0–1 | 78.3 [76.2,80.2] |
| 3 | 84.6 [81.7,87.1] | 61.5 [58.2,64.7] | 2 | 37.7 [35.4,40.1] |
| 4 | 66.9 [64.0,69.6] | 39.7 [36.7,42.7] | 3 | 29.7 [27.7,31.7] |
| 5 | 61.8 [58.9,64.7] | 35.0 [32.2,38.0] | 4+ | 15.9 [14.8,17.1] |
| 6+ | 45.2 [43.3,47.1] | 22.6 [21.0,24.3] | | |
| Number of sons | | | Number of sons | |
| 0–1 | 81.1 [79.2,82.7] | 64.7 [62.6,66.8] | 0 | 60.9 [58.6,63.2] |
| 2 | 66.7 [64.3,69.0] | 42.4 [40.1,44.8] | 1 | 35.1 [33.3,36.9] |
| 3 | 56.5 [53.9,59.0] | 32.8 [30.3,35.3] | 2 | 23.9 [22.3,25.6] |
| 4+ | 46.9 [44.4,49.3] | 23.6 [21.6,25.8] | 3+ | 17.2 [15.9,18.6] |
| Number of daughters | | | Number of daughters | |
| 0–1 | 81.0 [79.3,82.6] | 63.5 [61.3,65.5] | 0 | 56.3 [54.2,58.3] |
| 2 | 65.2 [62.7,67.5] | 43.8 [41.4,46.2] | 1 | 33.2 [31.5,35.0] |
| 3 | 56.9 [54.1,59.6] | 32.4 [30.0,34.9] | 2 | 24.4 [22.6,26.2] |
| 4+ | 46.2 [43.8,48.6] | 24.5 [22.5,26.7] | 3+ | 16.8 [15.2,18.6] |
| Sex composition | | | Sex composition | |
| 0–3 children | 89.3 [87.6,90.7] | 72.7 [70.6,74.6] | 0–1 | 78.3 [76.2,80.2] |
| 2 sons 2 daughters | 67.5 [62.5,72.0] | 38.7 [33.4,44.3] | 1 son 1 daughter | 33.7 [30.4,37.1] |
| 4 & sons>daughters | 67.0 [61.3,72.3] | 38.8 [33.3,44.5] | 2 sons 0 daughter | 41.1 [37.2,45.1] |
| 4 & daughters>sons | 66.3 [61.4,70.9] | 41.2 [36.4,46.1] | 0 sons 2 daughter | 44.5 [39.1,50.1] |
| 5 & sons> daughters | 61.0 [56.8,65.1] | 35.3 [31.5,39.3] | 3 & son>daughter | 27.5 [25.1,30.0] |
| 5 & daughters>sons | 62.7 [58.6,66.6] | 34.7 [30.8,38.9] | 3 & daughter>son | 32.2 [29.2,35.5] |
| 6+ children | 45.2 [43.3,47.1] | 22.6 [21.0,24.3] | 4+ | 15.9 [14.8,17.1] |
| Education | | | Education | |
| None | 65.2 [63.7,66.8] | 40.4 [38.3,42.6] | None | 29.6 [28.0,31.2] |
| Primary | 62.2 [59.5,64.8] | 43.0 [41.3,44.6] | Primary | 30.7 [29.2,32.2] |
| Secondary + | 62.8 [59.8,65.6] | 49.7 [47.2,52.2] | Secondary + | 33.3 [31.2,35.5] |
| Marital status | | | Marital status | |
| Never in a union | 73.8 [67.2,79.5] | 50.0 [44.8,55.3] | Never in a union | 50.8 [45.5,56.1] |
| Currently in a union | 63.1 [61.8,64.5] | 41.1 [39.7,42.5] | Currently in a union | 29.2 [28.2,30.2] |
| Formerly in a union | 66.7 [63.8,69.5] | 51.1 [48.8,53.4] | Formerly in a union | 39.8 [35.6,44.2] |
| Contraceptive use | | | Contraceptive use | |
| None | 66.6 [65.3,67.9] | 48.9 [47.5,50.3] | None | 39.0 [37.7,40.4] |
| Traditional | 47.6 [38.1,57.3] | 28.6 [23.3,34.7] | Traditional | 20.7 [18.3,23.4] |
| Modern | 44.6 [40.8,48.4] | 35.5 [33.4,37.6] | Modern | 17.5 [16.1,18.9] |
| Age at first birth | | | Age at first birth | |
| Less than 20 | 60.5 [58.7,62.3] | 40.3 [38.6,42.0] | Less than 20 | 20.5 [19.2,21.9] |
| 20–24 | 60.9 [58.6,63.2] | 41.1 [39.2,42.9] | 20–24 | 26.7 [25.2,28.2] |
| 25–29 | 69.3 [66.1,72.2] | 49.3 [45.7,53.0] | 25–29 | 39.2 [36.6,41.9] |
| 30–49 | 79.3 [75.5,82.7] | 71.0 [65.5,76.0] | 30–49 | 57.6 [54.0,61.1] |
| Exposure to FP messages | | | Exposure to FP messages | |
| Exposed | 61.8 [59.6,64.0] | 44.2 [42.5,45.9] | Exposed | 31.4 [29.7,33.1] |
| Not exposed | 65.5 [64.0,66.9] | 44.1 [42.4,45.8] | Not exposed | 30.8 [29.5,32.0] |
| Place of residence | | | Place of residence | |
| Urban | 61.9 [59.8,64.0] | 48.6 [46.2,51.1] | Urban | 28.3 [26.6,30.0] |
| Rural | 66.0 [64.6,67.5] | 42.4 [41.1,43.7] | Rural | 32.5 [31.3,33.8] |

(*Continued*)

**Table 3.** (Continued)

| Variable | Western and Central Africa | Eastern and Southern Africa | | South and Southeast Asia |
|---|---|---|---|---|
| Wealth Index | | | Wealth Index | |
| Lowest | 68.7 [66.4,70.9] | 43.5 [40.9,46.2] | Lowest | 33.5 [31.2,35.8] |
| Second | 66.1 [63.8,68.3] | 43.9 [41.3,46.5] | Second | 29.3 [27.6,31.1] |
| Middle | 61.9 [59.5,64.3] | 43.1 [40.5,45.7] | Middle | 30.5 [28.5,32.5] |
| Fourth | 62.1 [59.4,64.8] | 41.6 [39.0,44.3] | Fourth | 29.7 [27.9,31.7] |
| Highest | 61.7 [58.5,64.8] | 48.5 [45.7,51.3] | Highest | 32.2 [29.5,35.1] |

currently in a union. There was higher unrealized fertility among women who did not use any contraceptive method, but no large differences by exposure to FP messages. Unrealized fertility also increased with increased age at first birth with very high percentages of unrealized fertility among women whose age at first birth was between 30–49. Unrealized fertility was higher in rural areas in Western and Central Africa and South and Southeast Asia, and higher in urban areas for Eastern and Southern Africa, although the difference was not significant. There was no clear pattern between unrealized fertility and the wealth quintile. In Western and Central Africa, unrealized fertility was highest for women in the lowest wealth quintile, but in Eastern and Southern Africa, it was highest in the highest wealth quintile.

## Regression of unrealized fertility

Adjusted logistic regressions were fit for unrealized fertility with fertility related variables, socio-demographic variables, contraceptive use, and exposure to family planning messages. Figs 5 to 7 summarize the adjusted coefficients of the fertility related variables from each model. As shown in Table 2, three separate models were fit for the fertility related variables. Each model included women's education, marital status, age at first birth, wealth quintile, place of residence, contraceptive use, exposure to family planning messages, and country. The country variable used to control for between country variability in the models and their coefficients are not shown. The coefficients for the remaining variables are shown in Table 4. For these variables, the coefficients were very similar between the three models, and only the coefficients for Model 1 are shown.

The results for the two African regions shown in Figs 5 and 6 were very similar. In both regions, women with less than 4 children had a higher likelihood of unrealized fertility, while women who had more than 4 children had a lower likelihood of unrealized fertility compared to women with 4 children (see Model 1). In Model 2, the number of sons and daughters of a woman was examined. Compared to women with 2 sons or 2 daughters, women who had zero or one son or daughter had a higher likelihood of unrealized fertility and women with 3 or more sons or daughters had a lower likelihood of unrealized fertility. The coefficients for number of sons and number of daughters were almost identical. Model 3 includes the sex composition variable in which having 4 or 5 children was divided into categories of having more sons than daughters or more daughters than sons. For both regions, there appears to be little evidence of sex preferences. For women with 4 children, there was no significant difference in unrealized fertility if women had more sons than daughters or more daughters than sons compared to women with 2 sons and 2 daughters. For women with 5 children, there was significantly lower unrealized fertility, compared to women with 2 sons and 2 daughters, although the coefficients for the two categories were almost identical. This indicates that the lower likelihood of unrealized fertility was due to the number of children, as shown in Model 1, and not due to the number of sons or daughters of the women. In general, the results show that

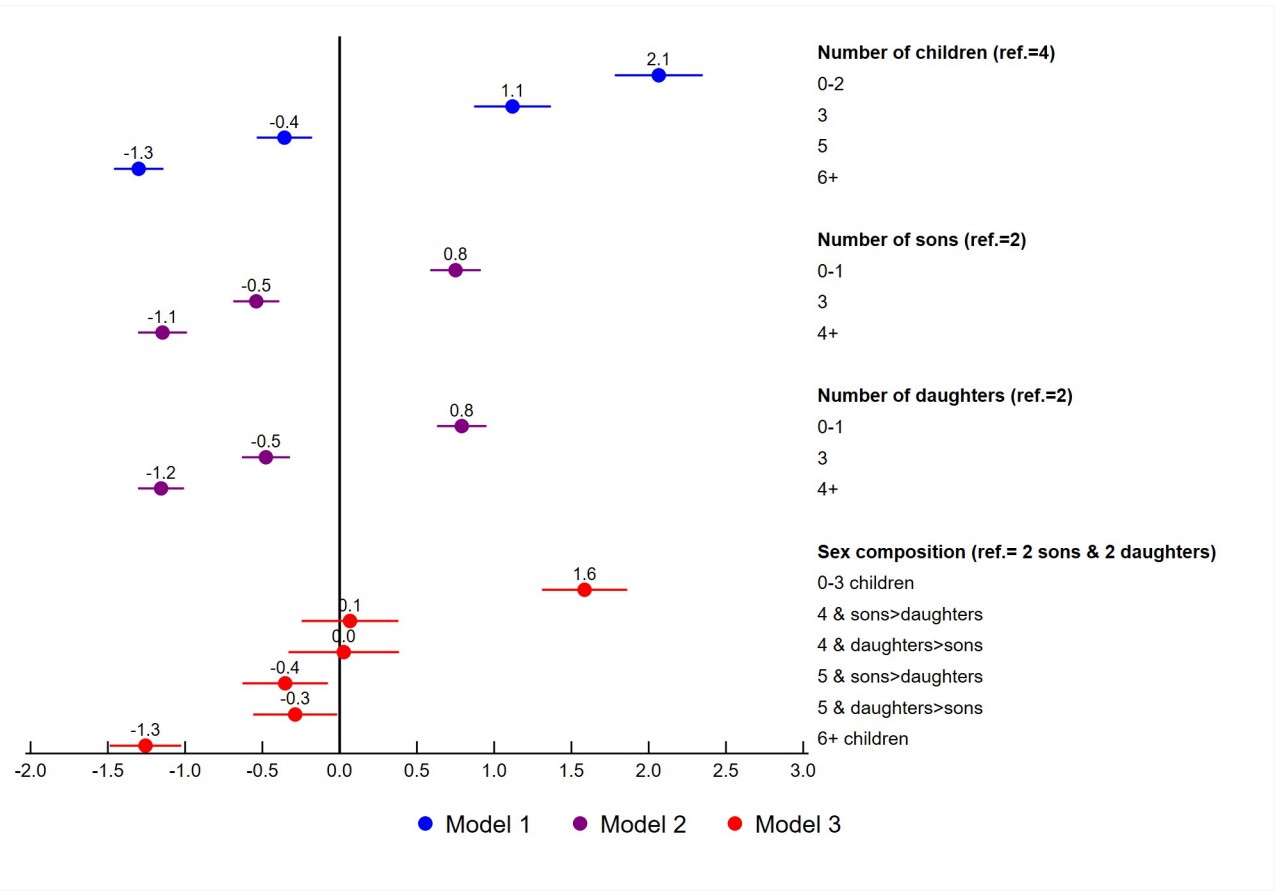

**Fig 5. Adjusted regression coefficients for the fertility variables among women age 44–48, Western and Central Africa.**

women from the two African regions prefer to have 4 children and more specifically, 2 sons and 2 daughters.

Fig 7 summarizes the adjusted coefficient results for the fertility variables for South and Southeast Asia. Since the distribution of the fertility variables differed by region, these variables were coded differently depending on the region as shown in Table 2. In Model 1, we see that compared to women who have 2 children, women with zero or 1 child have a higher likelihood of unrealized fertility, while women with 3 or more children have less likelihood of unrealized fertility. In Model 2, compared to women with 1 son or 1 daughter, women with no sons or no daughters were more likely to have unrealized fertility and women with 2 or more sons or daughters had a lower likelihood of unrealized fertility.

In Model 3, we examine the sex composition compared to women with 1 son and 1 daughter. In general, as with the results from the African regions, we see little evidence of sex preference and more important, to the number of children. Among women with 2 children, those with only sons or only daughters were more likely to have unrealized fertility compared to women with 1 son and 1 daughter. This indicates that women prefer to have 1 of each. The coefficients for women with only 2 sons or only 2 daughters in comparison to women with 1 son and 1 daughter were similar. Fig 7 also shows that women with 3 children, regardless of the sex composition, were less likely to have unrealized fertility compared to women with 1 son and 1 daughter. In summary, the results indicate that women in South and Southeast Asia prefer to have 2 children and specifically, 1 daughter and 1 son.

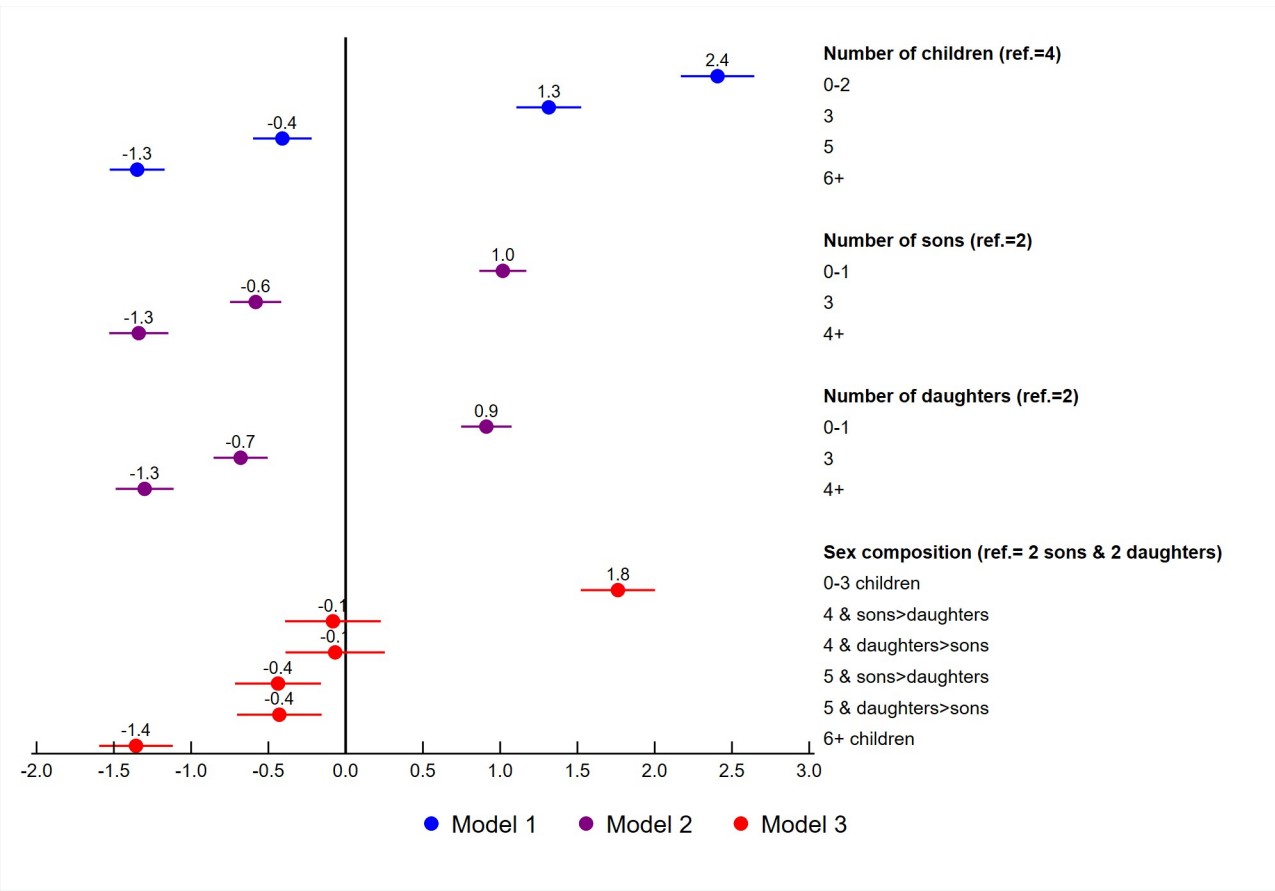

**Fig 6. Adjusted regression coefficients for the fertility variables among women age 44–48, Eastern and Southern Africa.**

Table 4 summarizes the coefficients for the remaining variables in the model for all regions. While the regression results for the fertility variables were consistent with the results of the crosstabulations with unrealized fertility in Table 3, this was not always the case for the remaining variables. While in Table 3 we observe an increase in unrealized fertility with increasing education in Eastern and Southern African and South and Southeast Asia, the opposite trend was observed after adjusting for other variables. In Table 4, for all three regions, the likelihood of unrealized fertility decreased with increasing education. Women with higher education were less likely to have unrealized fertility. In addition, crosstabulations have shown higher unrealized fertility for women not currently in a union (both never and formally in a union), although in Table 4 we see an opposite trend.

For all regions, women who were never in a union had a lower likelihood of unrealized fertility compared to women currently in a union. This was also the case for women formerly in a union in the two African regions although with a smaller difference. Consistent with the findings from the crosstabulations are the results for the wealth quintile. For Western and Central Africa, we observe a decrease in unrealized fertility with increasing wealth quintile, but with no difference between the second and lowest wealth quintile. In South and Southeast Asia, only women from the second and fourth wealth quintiles had a significantly lower likelihood of unrealized fertility compared to women from the lowest wealth quintile. In addition, for all regions, women who use either traditional or modern contraceptive methods had a lower likelihood of unrealized fertility compared to women who did not use any method. However,

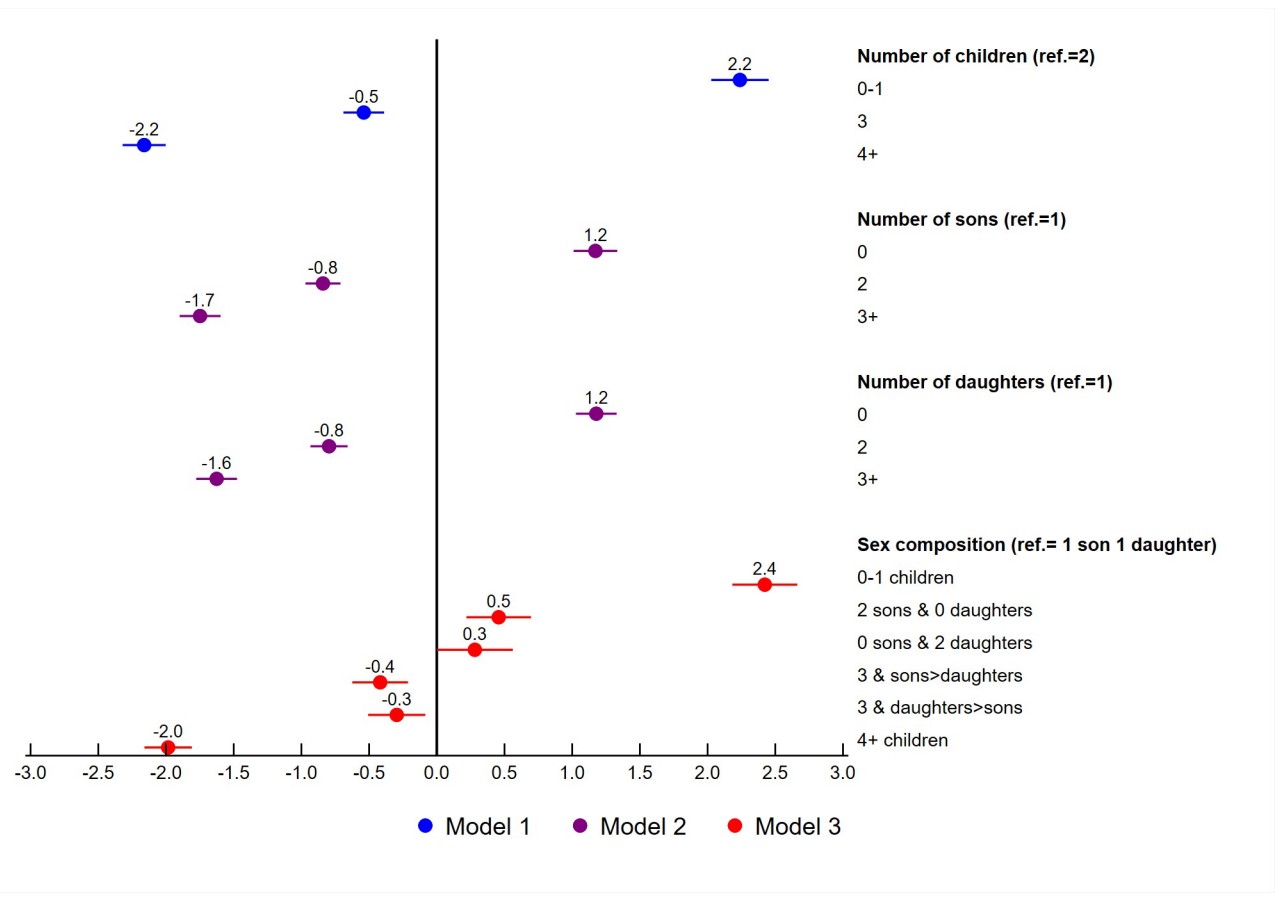

**Fig 7. Adjusted regression coefficients for the fertility variables among women age 44–48, South and Southeast Asia.**

exposure to family planning messages was not significantly associated with unrealized fertility. In Eastern and Southern African and South and Southeast Asia, women who lived in rural areas had a higher likelihood of unrealized fertility compared to women from urban areas. Finally, while there were large differences in unrealized fertility and age at first birth in Table 3, this did not appear to be an important predictor of unrealized fertility among women age 44–48 according to the regression models.

## Discussion

Our findings reveal some distinctions among regions, particularly between SSA and South and Southeast Asia. This study indicated that, when compared to the other regions, West and Central Africa, followed by Eastern and Southern Africa, have the greatest mean ideal number of children and the largest proportions of unrealized fertility. While fertility preferences are high in SSA, there appears to be a limit to this ideal. In general, women from the African regions desire 4 children, while women from South and Southeast Asia desire 2 children. Deviations from this ideal have significant changes in unrealized fertility.

Our findings highlight that more women in the African regions have unrealized fertility compared to women in South and Southeast Asia. Specifically, women in Western and Central Africa have both the highest percent of unrealized fertility and the highest mean ideal number of children. Exemplifying this trend is Chad, which has both the highest ideal number of

**Table 4. Adjusted coefficients of the control variables in Model 1 from the logistic regressions of unrealized fertility.**

| Variable | Western and Central Africa | Eastern and Southern Africa | South and Southeast Asia |
|---|---|---|---|
| Education (ref. = None) | | | |
| Primary | -0.17* | -0.17* | -0.27*** |
| Secondary+ | -0.51*** | -0.43*** | -0.34*** |
| Marital status (ref. = Currently in a union) | | | |
| Never in a union | -0.56* | -0.52** | -1.54*** |
| Formerly in a union | -0.25** | -0.19** | 0.02 |
| Age at first birth (ref. = Less than 20) | | | |
| 20–24 | -0.05 | -0.12 | -0.01 |
| 25–29 | -0.03 | -0.09 | 0.11 |
| 30–49 | -0.20 | 0.18 | -0.02 |
| Wealth quintile (ref. = Lowest) | | | |
| Second | -0.14 | 0.02 | -0.20* |
| Middle | -0.43*** | -0.05 | -0.17 |
| Fourth | -0.50*** | -0.18 | -0.20* |
| Highest | -0.79*** | -0.25 | -0.20 |
| Place of residence (ref. = Urban) | | | |
| Rural | 0.12 | 0.25** | 0.25** |
| Contraceptive use (ref. = None) | | | |
| Traditional | -0.49* | -0.46** | -0.35*** |
| Modern | -0.51*** | -0.44*** | -0.45*** |
| Exposure to FP messages (ref. = Exposed) | | | |
| Not exposed | -0.05 | 0.12 | -0.04 |

Note:

*$p<0.05$,

**$p<0.01$,

***$p<0.001$

children (9.1) and unrealized fertility (76.8%). Unwanted fertility was highest in the regions of Eastern and Southern Africa and South and Southeast Asia. Notably, countries in this region, such as Bangladesh and Nepal, have both the highest rates of unwanted fertility (53 and 59%, respectively) and are among the countries with the lowest mean ideal number of children (2.6 and 2.5, respectively). The results have shown that overall, women in non-SSA regions want to reach replacement level fertility. Women in these regions ideally want two children. Several countries in these regions have high percentages of women with unwanted fertility, as well as high percentages of women reaching their ideal number (see S3 Table). In the two African regions, the ideal number of children was double or more than double the replacement level. Women in Africa want large families even in later stages of fertility and after having several children. Fertility and family planning programs should consider these differences in regional patterns.

In examining the distribution in unwanted and unrealized fertility by sex of the child, the findings suggest that the number of children is of greater importance than the sex of the children. Unrealized fertility decreased as the number of children, sons, and daughters increased. Overall, there are minimal differences in unrealized fertility for boys versus girls. Across all regions, the total percentage of unrealized fertility is consistently higher than the unrealized fertility for boys or girls. However, there were some regional and country differences. For

example, in South and Southeast Asia, a slight preference for boys is observed and this is mainly driven by the higher levels of unrealized fertility for boys in Afghanistan and Pakistan compared to girls (see S3 Table).

Although there were some regional trends, the findings also highlight some country-specific deviations. Since there is a strong association between reproductive circumstances and fertility preferences [10], the deviating country specific findings could be a result of a country's unique sociocultural or socioeconomic context. Findings from the regression analysis also highlight some noteworthy socioeconomic factors that affect unrealized fertility. Women from high wealth quintiles decreased their likelihood of unrealized fertility compared to the lowest quintile, which indicated that economic constraints might be a factor in unrealized fertility. This could imply that wealthier women were able to have or exceed their ideal number of children compared to women from the lowest wealth quintile.

Studies suggest that fertility behavior is influenced by socioeconomic factors and changes in both macro and micro economic factors [14, 15]. Further, women from higher wealth quintiles are better able to mitigate their fertility intentions through delayed marriage and modest increases in contraceptive use [11]. Our findings show that women who use either traditional or modern contraceptive methods also have a lower likelihood of unrealized fertility. Further, place of residence could also be a factor since women in rural areas have a higher likelihood of unrealized fertility than women in urban areas. As noted, access to resources based on place of residence can be a driving factor.

We observed in the regression results for all three regions that unrealized fertility decreased with increasing education. These results did not follow the trends in the crosstabulations that show that unrealized fertility increased with increasing education in two regions. The adjustment for the other variables in the model has modified the relationship of the education variable with unrealized fertility. Further analysis of this finding has shown that it was due to the addition of the number of children variable in the model. Several studies have found that women from lower education levels have higher unwanted fertility and pregnancies [8, 12, 13, 16]. The results that show women with lower education also have higher unrealized fertility appears to be driven by the high fertility preferences and higher number of children for women with lower education.

Analysis of DHS surveys from 58 countries found that mean ideal number of children decreases with increasing education level [7]. However, the same study found no consistent relationship between unrealized fertility and education, and high unrealized fertility across all educational levels. In this analysis, we found a marginal difference in the African regions between women with no education and women with primary education in unrealized fertility, while in the South and Southeast Asia region, there were very similar levels of lower unrealized fertility for women with primary and secondary or more education compared to women with no education (see Table 4). Therefore, we observe again that programs must consider these regional patterns and differences in sociodemographic variables.

Other factors not examined in this study could also be linked to the partner's desires, community factors, or other hidden variables. For example, some women could be postponing childbirth due to individual factors not measured in DHS data. This postponement could also lead to women never reaching the ideal. This study focuses on women age 44–48 who are more likely to have higher levels of infertility. A study of 277 countries with DHS data found that secondary infertility (the inability to have a child after having one or more) ranged from 4% to 22% [17], and several studies have found that women of advanced age were more likely to experience infertility [18]. Therefore, postponement for various reasons could lead to more difficulty in reaching the ideal number of children. Women could also be reporting an ideal they do not expect to achieve. Postponement of childbirth or the decision to remain childless

were found to be linked to women's prioritizing education and their careers, the search for the right partner, and the absence of supportive family policies [19–23].

Limitations in this analysis begin with the question on the ideal number of children. Women could be rationalizing their response to this question by reporting the number of children they have as their ideal. Rationalization is also expected to increase with increasing age and when asked about sex specific ideals. This could explain the limited evidence of sex preferences. Another limitation when examining sex preferences is the numerous possible combinations of sons and daughters a woman can have. Extreme distributions, such as couples who have all sons or all daughters, have sample size limitations and were therefore not included in the analysis. In addition, the likelihood of wanting to have a child with a different sex than the one(s) you have is very high and possibly not due to sex preferences.

Another bias can be due to the large percentage of non-numeric responses for ideal number of children in some countries. Women who gave a non-numeric response were excluded from the analysis and this exclusion could have some effect on the findings and for some regions more than others. In addition, providing non-numeric responses has been found to be significantly and negatively associated with education and knowledge and use of modern contraceptive methods [24]. Therefore, it is not likely to be equally distributed across women within each country. On the other hand, one can argue, as cited by Casterline & Han (2017), that the measure of unrealized fertility by design requires positive evidence of wanting more children. Finally, the analysis did not consider the fertility preferences of the women's husband. The husband's preferences can have an effect on fertility behaviors in a couple and can therefore be an important factor that should be explored.

Women should have the right to have the number of children they want. However, population and reproductive programs in LMICs generally promote smaller family sizes through family planning (FP) to reduce fertility and improve women's and child health outcomes. Increases in contraceptive use not only reduce unwanted pregnancies, high-risk pregnancies, and maternal and infant mortality, but also have been found to improve schooling and economic outcomes, both at the individual and macro levels [6]. Our results found that exposure to FP messages was not significantly associated with unrealized fertility. Perhaps this can be an opportunity for developing FP messages that target women with high unrealized fertility to promote women's control over their reproductive health. Regional differences and country specific deviations should also be considered. For example, FP programs and messages cannot promote the same number of children in all regions or countries, because this could be drastically different from the ideal of most women. Cultural and socioeconomic contexts should also be considered for more effective and targeted interventions.

## Supporting information

**S1 Table. Among women age 15–49, the percentage of women who had a birth between age 44–48.**
(XLSX)

**S2 Table. Percent distribution of ideal number of children and mean ideal number of children for women age 44–48.**
(XLSX)

**S3 Table. Percent distribution with 95% confidence intervals of ideal minus current number of children (total, boys, girls) among women 44–48.**
(XLSX)

## Acknowledgments

The authors wish to thank Sara Yeatman and Thomas Pullum for their comments on an earlier version of this work. We would also like to thank Diane Stoy for editing the paper.

## Author Contributions

**Conceptualization:** Shireen Assaf.

**Formal analysis:** Shireen Assaf.

**Methodology:** Shireen Assaf.

**Supervision:** Shireen Assaf.

**Writing – original draft:** Shireen Assaf, Lwendo Moonzwe Davis.

**Writing – review & editing:** Shireen Assaf, Lwendo Moonzwe Davis.

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
