## [Decision Letter · Decision Letter 0]

8 Jun 2022

PONE-D-21-30775Unrealized fertility among women in low- and middle-income countries with Demographic and Health Surveys dataPLOS ONE

Dear Dr. Assaf,

Thank you for submitting your manuscript to PLOS ONE. After careful consideration, we feel that it has merit but does not fully meet PLOS ONE’s publication criteria as it currently stands. Therefore, we invite you to submit a revised version of the manuscript that addresses the points raised during the review process.

The manuscript has been evaluated by a reviewer, and their comments are available below.

The reviewer raised a number of concerns that need attention. They request further information to strengthen the rationale for examining associations described in the manuscript, and the analytical measures and approaches adopted. The reviewer also requests copyediting of the manuscript to improve language and clarity.

Could you please revise the manuscript to carefully address the concerns raised?

Please note that we have only been able to secure a single reviewer to assess your manuscript. We are issuing a decision on your manuscript at this point to prevent further delays in the evaluation of your manuscript. Please be aware that the editor who handles your revised manuscript might find it necessary to invite additional reviewers to assess this work once the revised manuscript is submitted. However, we will aim to proceed on the basis of this single review if possible.

We look forward to receiving your revised manuscript.

Kind regards,

Jamie Royle

Staff Editor

PLOS ONE

Journal Requirements:

Reviewers' comments:

Reviewer's Responses to Questions

**Comments to the Author**

1. Is the manuscript technically sound, and do the data support the conclusions?

Reviewer #1: Yes

2. Has the statistical analysis been performed appropriately and rigorously? 

Reviewer #1: Yes

3. Have the authors made all data underlying the findings in their manuscript fully available?

Reviewer #1: Yes

4. Is the manuscript presented in an intelligible fashion and written in standard English?

Reviewer #1: No

5. Review Comments to the Author

Reviewer #1: This multi-country, multi-region study on the recent level of unrealized fertility (surveys since 2014) and associations with sex preference and composition is a useful contribution to a rather limited evidence base on the gap between family size desires and actual experiences.

Suggestions for revision below are primarily to strengthen the rationale for examining particular associations and the analytical measure and approach adopted.

1. That LAC is represented by only three countries (Colombia, Guatemala and Haiti) severely limits the interpretation and meaningfulness of aggregated “regional” results throughout the paper. Suggest replacing figures 2-6 with a country-specific bar chart similar to figure 1 that shows the recent level of unrealized fertility (because it is not clear what the purpose is to compare the distribution of unwanted fertility, exact match and unrealized fertility by regional level).

2. Motivate and state each factor more clearly (e.g., at a minimum Casterline and Ha ask “are women with fewer children more likely to have fallen short of their childbearing goals?).

Education is used as a “control variable” for unrealized fertility but described as a result in the abstract and body of the paper. Authors should hypothesize and interpret education as a key association (e.g., Channon and Harper note “We would expect, a priori, that more highly educated individuals would be better equipped to translate their preferences correctly.”). This may be more challenging when examining contemporaneous characteristics (contraceptive use, exposure to FP messages).

Given that this is a recent cross-sectional examination of unrealized fertility with one measure, authors could examine what other associations may make sense theoretically. For example, are women who are divorced or widowed more likely to have unrealized fertility (e.g., other relationship factors derailed achieving their family size desires)?

3. Abstract conclusion and paper’s introduction (first para): What is a “healthy family size” that family planning programs and messages are meant to promote? Authors could apply their initial reproductive rights rationale: that the level of unrealized fertility and associations by education and wealth quintile must be accounted for in family planning messaging—that not all people have neared the end of their reproductive years reaching the family size they wanted.

4. Why use a wider age group (40-49) than the near end of fecund years (e.g., 44-48)? Even though the authors note the small percentage of women in this age group who have a birth, it is not evenly distributed across the 10-year age group. Note: Casterline and Ha used the age group 44-48 (omitting 49 because of age displacement in survey responses).

Minor points

5. Be clear in the Introduction that the study focuses on the quantum aspect of unrealized fertility (falling short of the desired number of children) and does not consider the timing aspect (having births later than intended), and that the study the examines one recent time point compared to published estimates from earlier years (including time trends).

6. Introduction (para 2) Recommend bringing in findings on level from Casterline and Han as well, and note how different measures likely serve as upper and lower bounds (and that the measure used in this analysis—ideal family size greater than number of living children—could be considered an upper bound).

7. Authors acknowledge “Women who provided a nonnumerical response were excluded from the analysis. This is one of the limitations of our analysis that would disproportionally affect some regions more than others that have higher percentages of nonnumerical responses.” Countries with relatively high levels are noted on page 6. Could bring back into the discussion more than just the limitation of omitting women with non-numeric preferences from the analysis but also touch on the interconnected links – that non-numeric fertility preferences are especially influenced by educational attainment and knowledge of contraception (see Frye M, Bachan L. The demography of words: The global decline in non-numeric fertility preferences, 1993–2011. Population Studies. 2017; 71(2):187–209. https://doi.org/10.1080/00324728.2017)).

8. Copy editing needed throughout the manuscript (e.g., missing articles, spelling, etc.)

6. PLOS authors have the option to publish the peer review history of their article (what does this mean?). If published, this will include your full peer review and any attached files.

Reviewer #1: **Yes: **Ann Biddlecom

---

## [Author Response · Author response to Decision Letter 0]

22 Jul 2022

We would like to thank the reviewer for these useful comments. We provide a point-by-point response below.

Thank you.

Comments:

Reviewer #1: This multi-country, multi-region study on the recent level of unrealized fertility (surveys since 2014) and associations with sex preference and composition is a useful contribution to a rather limited evidence base on the gap between family size desires and actual experiences.

Suggestions for revision below are primarily to strengthen the rationale for examining particular associations and the analytical measure and approach adopted.

1. That LAC is represented by only three countries (Colombia, Guatemala and Haiti) severely limits the interpretation and meaningfulness of aggregated “regional” results throughout the paper. Suggest replacing figures 2-6 with a country-specific bar chart similar to figure 1 that shows the recent level of unrealized fertility (because it is not clear what the purpose is to compare the distribution of unwanted fertility, exact match and unrealized fertility by regional level).

Response: We agree that the representation of LAC region by only three countries is very limited. This is also the case for the North Africa and West and Central Asia region with five diverse countries. There is relatively lower availability of DHS data in these regions compared to the remaining regions. We had mentioned this as part of the limitations to this paper; however, revisiting this and based on your comment we believe it is best to only focus on the two African regions and South and Southeast Asia as these three regions are represented by a larger number of countries. We have also updated two countries that have had more recent surveys since this revision (Rwanda and India) and added another country to the analysis that had data released after the revision (Mauritania). The updated country list is in Table 1. 

2. Motivate and state each factor more clearly (e.g., at a minimum Casterline and Ha ask “are women with fewer children more likely to have fallen short of their childbearing goals?).

Education is used as a “control variable” for unrealized fertility but described as a result in the abstract and body of the paper. Authors should hypothesize and interpret education as a key association (e.g., Channon and Harper note “We would expect, a priori, that more highly educated individuals would be better equipped to translate their preferences correctly.”). This may be more challenging when examining contemporaneous characteristics (contraceptive use, exposure to FP messages).

Given that this is a recent cross-sectional examination of unrealized fertility with one measure, authors could examine what other associations may make sense theoretically. For example, are women who are divorced or widowed more likely to have unrealized fertility (e.g., other relationship factors derailed achieving their family size desires)?

Response: We added our hypothesis for the factors association with unrealized fertility at the end of the introduction. Also, we agree that education has direct links to the outcome and is not only a control. The language in the paper was corrected to reflect this in the abstract and methods. We have also added marital status as an independent variable and the findings were added to the results. 

3. Abstract conclusion and paper’s introduction (first para): What is a “healthy family size” that family planning programs and messages are meant to promote? Authors could apply their initial reproductive rights rationale: that the level of unrealized fertility and associations by education and wealth quintile must be accounted for in family planning messaging—that not all people have neared the end of their reproductive years reaching the family size they wanted.

Response: Agree. We should have framed this in a reproductive right lens. This was corrected. 

4. Why use a wider age group (40-49) than the near end of fecund years (e.g., 44-48)? Even though the authors note the small percentage of women in this age group who have a birth, it is not evenly distributed across the 10-year age group. Note: Casterline and Ha used the age group 44-48 (omitting 49 because of age displacement in survey responses).

Response: We originally selected 40-49 since we found small percentages of births in this age group and also to increase our sample size. We have now limited the analysis to women 44-48 as suggested. 

Minor points

5. Be clear in the Introduction that the study focuses on the quantum aspect of unrealized fertility (falling short of the desired number of children) and does not consider the timing aspect (having births later than intended), and that the study the examines one recent time point compared to published estimates from earlier years (including time trends).

Response: This was added to the last paragraph of the introduction. 

6. Introduction (para 2) Recommend bringing in findings on level from Casterline and Han as well, and note how different measures likely serve as upper and lower bounds (and that the measure used in this analysis—ideal family size greater than number of living children—could be considered an upper bound).

Response: Thank you for pointing this out. We added this to the last paragraph of the introduction that discuses what the paper will cover. 

7. Authors acknowledge “Women who provided a nonnumerical response were excluded from the analysis. This is one of the limitations of our analysis that would disproportionally affect some regions more than others that have higher percentages of nonnumerical responses.” Countries with relatively high levels are noted on page 6. Could bring back into the discussion more than just the limitation of omitting women with non-numeric preferences from the analysis but also touch on the interconnected links – that non-numeric fertility preferences are especially influenced by educational attainment and knowledge of contraception (see Frye M, Bachan L. The demography of words: The global decline in non-numeric fertility preferences, 1993–2011. Population Studies. 2017; 71(2):187–209. https://doi.org/10.1080/00324728.2017)).

Response: Thank you for providing this reference. More was added on this limitation on page 16. 

8. Copy editing needed throughout the manuscript (e.g., missing articles, spelling, etc.)

Response: This revised paper has been edited by a professional editor and references were checked.

---

## [Editor Report · Decision Letter 1]

11 Oct 2022

Unrealized fertility among women in low- and middle-income countries

PONE-D-21-30775R1

Dear Dr. Assaf,

We’re pleased to inform you that your manuscript has been judged scientifically suitable for publication and will be formally accepted for publication once it meets all outstanding technical requirements.

Kind regards,

Dorina Onoya

Academic Editor

PLOS ONE